# Synergizing Success: The Role of Anlotinib Combinations in Advanced Non-Small Cell Lung Cancer Treatment

**DOI:** 10.3390/ph18040585

**Published:** 2025-04-16

**Authors:** Helal F. Hetta, Hashim M. Aljohani, Nizar Sirag, Hassabelrasoul Elfadil, Ayman Salama, Rand Al-Twalhy, Danah Alanazi, Manal D. Al-johani, Jumanah H. Albalawi, Rinad M. Al-Otaibi, Raghad A. Alsharif, Reem Sayad

**Affiliations:** 1Division of Microbiology, Immunology and Biotechnology, Department of Natural Products and Alternative Medicine, Faculty of Pharmacy, University of Tabuk, Tabuk 71491, Saudi Arabia; habdelgadir@ut.edu.sa; 2Department of Clinical Laboratory Sciences, College of Applied Medical Sciences, Taibah University, Madina 41477, Saudi Arabia; hsnani@taibahu.edu.sa; 3Department of Pathology and Laboratory Medicine, College of Medicine, University of Cincinnati, Cincinnati, OH 45221, USA; 4Division of Pharmacognosy, Department of Natural Products and Alternative Medicine, Faculty of Pharmacy, University of Tabuk, Tabuk 71491, Saudi Arabia; nizarsirag@gmail.com; 5Department of Pharmaceutics, Faculty of Pharmacy, University of Tabuk, Tabuk 71491, Saudi Arabia; agrawan@ut.edu.sa; 6PharmD Program, Faculty of Pharmacy, University of Tabuk, Tabuk 71491, Saudi Arabia; 421001509@stu.ut.edu.sa (R.A.-T.); 421000390@stu.ut.edu.sa (D.A.); mnal87038@gmail.com (M.D.A.-j.); 421000067@stu.ut.edu.sa (J.H.A.); 421001053@stu.ut.edu.sa (R.M.A.-O.); 421000501@stu.ut.edu.sa (R.A.A.); 7Department of Histology, Faculty of Medicine, Assiut University, Assiut 71515, Egypt; reem.17289806@med.aun.edu.eg

**Keywords:** non-small cell lung cancer(NSCLC), lung carcinoma, anlotinib, a novel multi-targeting tyrosine kinase inhibitor, novel targeted drug

## Abstract

Anlotinib, a novel receptor tyrosine kinase inhibitor that is taken orally, targets several RTKs and is authorized as a third-line treatment for patients with advanced non-small cell lung cancer (NSCLC). Anlotinib is also used in combination with immunotherapy or chemotherapy for advanced NSCLC. We aimed to explore the efficacy and safety of anlotinib-based regimens in NSCLC treatment, focusing on combination therapies. We also addressed challenges that hinder oncologists from using it, such as toxicity and resistance mechanisms. A systematic approach involves searching the National Institute of Health PubMed, Scopus, MedLine, and Web of Science databases up to April 2024. Relevant studies were identified and analyzed for their methodologies, outcomes, and patient characteristics. Findings revealed that numerous effective combination regimens, such as anlotinib with platinum-based chemotherapy and anlotinib combined with PD-1 blockades, have shown positive results in terms of progression-free survival (PFS), overall survival (OS), and objective response rate (ORR). On the other hand, NSCLC treatment faces hurdles due to drug resistance and its toxicity profile. These challenges underscore the need for continued research and optimization of treatment strategies.

## 1. Introduction

Lung cancer is one of the most prevalent malignant tumors and is the leading cause of cancer-related death worldwide. About 238,340 persons in the US receive a lung cancer diagnosis each year. An estimated 127,070 patients die annually. More people are dying from lung cancer than from prostate, brain, colorectal, and breast cancers altogether [1]. In 2023, smoking cigarettes was the primary cause of around 103,000 (81%) of the 127,070 lung cancer deaths, with second-hand smoke accounting for an additional 3560 fatalities. The remaining balance of almost 20,500 lung cancer deaths not related to smoking would be ranked as the eighth most common cause of cancer fatalities in both sexes [2].

In 2021, the World Health Organization (WHO) categorized lung cancers based on immunohistochemistry and light microscopy to help predict a prognosis and better guide treatment. It is categorized into small-cell lung cancer and non-small-cell lung cancer (NSCLC) [3].

NSCLC accounts for approximately 85% of all lung cancer cases and remains the leading cause of cancer-related mortality worldwide. According to the Global Cancer Observatory (GLOBOCAN) 2020 data, lung cancer was the second most commonly diagnosed cancer (11.4% of total cases) and the leading cause of cancer death (18% of total cancer deaths) globally, with an estimated 2.2 million new cases and 1.8 million deaths annually [4]. Geographically, the burden of NSCLC varies significantly. The highest incidence rates are observed in Eastern Asia, particularly in China, which alone accounts for over one-third of global lung cancer cases and deaths [5]. Other regions with high incidence include Central and Eastern Europe and North America. Conversely, lower incidence rates are reported in parts of Africa and South-Central Asia, although underdiagnosis and limited access to healthcare may contribute to underestimation in these regions [6]. This geographical variation is influenced by multiple factors, including differences in smoking prevalence, environmental exposures (e.g., air pollution, radon), occupational hazards, and genetic susceptibility. Additionally, the increasing incidence of NSCLC among never-smokers, particularly in East Asian populations, underscores the complexity of its etiology [7].

NSCLC refers to a group of lung malignancies, including adenocarcinoma, squamous cell carcinoma (SSC), and large cell carcinoma [3]. Half of all cases of lung cancer are adenocarcinomas, which are the most prevalent type of NSCLC. Before this time, the most common type of NSCLC to be diagnosed with lung cancer was SSC. Although the tracheobronchial tree is often where SCC first appears, it can mostly be found near the lung’s periphery [8]. Large cell carcinoma is an exclusion diagnosis because of its poor differentiation. Electron microscopy and immunohistochemistry (IHC) cannot be used to detect it [9].

A controlled randomized trial found that positron emission tomography (PET) staging performed in conjunction with computed tomography (CT) was more accurate in classifying N-stage diagnoses than standard invasive staging methods like biopsy of the mediastinal lymph node with echo endoscopy [10,11]. Another controlled trial confirmed that any positive node on PET-CT has to be sampled [12]. For patients who receive treatment due to a curative goal for their condition or for those exhibiting symptoms or signs suggestive of brain metastasis, a magnetic resonance imaging (MRI) or CT of the head is advised. The new multidisciplinary classification of lung cancer by the American Thoracic Society, European Respiratory Society, and International Association for the Study of Lung Cancer states that obtaining enough tissue material is essential [13,14]. Initial evaluation of suspected cases and individualized treatment can be assessed by the identification of the mutation.

Treatment strategies for NSCLC vary based on the disease stage. While early-stage patients (stage I or II) may benefit from surgical resection or, in non-surgical candidates, conventional or stereotactic radiotherapy, the therapeutic landscape for advanced or refractory NSCLC is more complex and evolving. Patients with advanced NSCLC often require systemic therapies, including chemotherapy, targeted therapy, immunotherapy, or combinations thereof. In cases where surgery or radiotherapy fails or is not an option, minimally invasive modalities such as microwave ablation, cryoablation, and radiofrequency ablation have shown utility as palliative or salvage treatments [15,16,17].

A significant subset of advanced NSCLC patients presents with mutations, particularly in the epidermal growth factor receptor (EGFR). EGFR mutations are found in approximately 15% of NSCLC patients in Western populations and are more prevalent in Asian populations, non-smokers, and females [18,19]. EGFR-targeted therapies have become a cornerstone of management in these cases, with successive generations of EGFR tyrosine kinase inhibitors (TKIs)—including first-generation (erlotinib, gefitinib) [20,21], second-generation (afatinib, dacomitinib) [22,23], and third-generation agents (osimertinib) [24,25]—demonstrating substantial clinical benefits. Despite these advances, resistance to targeted therapies remains a major challenge in refractory NSCLC, highlighting the need for novel treatment strategies, such as combination regimens involving agents like anlotinib, to improve long-term outcomes.

Anlotinib, a novel receptor tyrosine kinase inhibitor that is taken orally, targets numerous RTKs, such as c-kit, PDGFR, FGFR, and VEGFR. In May 2018, the National Medical Products Administration (NMPA) authorized anlotinib as a third-line therapy for advanced NSCLC because of its great efficacy and few adverse effects in the ALTER 0303 study. Anlotinib significantly increased progression-free survival (PFS) with tolerable toxicity in subsequent lines of treatment for advanced NSCLC, according to the preliminary results of our real-world trial [26,27].

Anlotinib is used with chemotherapy and immunotherapy for advanced NSCLC. So, this study aims to comprehensively review and evaluate the therapeutic potential of anlotinib in the treatment of advanced NSCLC, with a particular focus on its efficacy as a monotherapy and in combination with various treatment modalities, including chemotherapy, radiotherapy, EGFR-TKIs, and immunotherapy.

## 2. Anlotinib Mechanism of Action in Advanced NSCLC

Anlotinib is a novel, orally administered small-molecule TKI with broad-spectrum anti-tumor activity. It exerts its therapeutic effects through the inhibition of multiple receptor tyrosine kinases (RTKs) involved in tumor angiogenesis, growth, and metastasis. It potently inhibits vascular endothelial growth factor receptors (VEGFR-1, VEGFR-2, and VEGFR-3), which play a central role in tumor vascularization. Inhibition of VEGFRs disrupts new blood vessel formation, thereby limiting oxygen and nutrient supply to tumor cells. Anlotinib also targets fibroblast growth factor receptors (FGFR1 to FGFR4), platelet-derived growth factor receptors (PDGFR-α and PDGFR-β), and c-Kit (stem cell factor receptor), which are involved in tumor proliferation, survival, and resistance mechanisms. By blocking VEGFR-3 and FGFR pathways, anlotinib may also inhibit lymphatic vessel formation and metastatic spread (Figure 1). This multi-targeted mechanism allows anlotinib to effectively suppress tumor progression through both antiangiogenic and antiproliferative effects. Its broad spectrum of RTK inhibition makes it particularly valuable in the treatment of advanced or refractory cancers, including NSCLC [28,29].

According to preclinical research, anlotinib binds to the ATP-binding pocket of VEGFR-2 tyrosine kinase and selectively inhibits VEGFR-2 (IC_50_ < 1 nmol/L), which in turn prevents human umbilical vein endothelial cells (HUVECs) from proliferation in response to VEGF. Additionally, anlotinib decreased vascular density in vivo and inhibited the migration of HUVEC, the formation of the tube, and the growth of microvessels in vitro. Anlotinib demonstrated superior and wider antitumor activity compared to sunitinib in vivo [28]. Anlotinib reduced the number of cells in cell lines harboring mutant FGFR2 proteins. However, the combination of anlotinib with paclitaxel and carboplatin did not seem to be more effective than anlotinib alone, as was the case with other oral RTK inhibitors [30].

Another preclinical trial reported that anlotinib suppresses the development of capillary-like tubes and cell migration in endothelial cells that are stimulated by VEGF/PDGF-BB/FGF-2.

Additionally, both in vitro and in vivo, anlotinib markedly inhibited the angiogenesis induced by VEGF/PDGF-BB/FGF-2. Anlotinib suppresses downstream ERK signaling as well as the activation of VEGFR2, PDGFR-β, and FGFR1. Two other anti-angiogenesis medications, nintedanib and sunitinib, have less anti-angiogenic activity than anlotinib [31].

## 3. Anlotinib Monotherapy: Clinical Insights and Therapeutic Efficacy

NSCLC is one of the most frequent diseases used to assess anlotinib’s safety and efficacy. Anlotinib has mostly been studied in advanced NSCLC since the third-line treatment of these patients is currently not standardized.

A randomized, multicenter study was conducted to evaluate anlotinib’s efficacy and safety as a monotherapy for patients with refractory NSCLC. A total of 117 patients were assigned to receive either anlotinib or placebo (12 mg per day, per os; days 1–14; 21 days each cycle). It was found that anlotinib prolongs the PFS of the patients in its group more than those of the placebo (4.8 vs. 1.2 months; *p * <  0.0001). Furthermore, the anlotinib group’s ORR was higher than the placebo group’s (10.0% vs. 0%; *p* = 0.028). Significantly, all subgroups, except for the one with three or fewer metastases, improved from anlotinib treatment regardless of age, sex, the effectiveness of prior treatments, stage, smoking history, or histology. Furthermore, anlotinib also prolongs the duration of the overall survival (OS) of patients in its group than those in the control group (9.3 vs. 6.3 months). The difference in OS between the two groups is not statistically significant (*p* = 0.2316). It may be due to the small sample size of both groups, which cannot detect the difference [27].

Another controlled phase III trial (ALTER-0303) assessed the safety and effectiveness of anlotinib in advanced NSCLC. A total of 437 participants were assigned to oral anlotinib (12 mg QD) or a placebo group (from days 1 to 14 of a 21-day cycle). Treatment was administered until the tumor progressed or it was stopped because of its toxicity. According to the findings, anlotinib is more effective than a placebo as a third-line treatment for individuals with advanced NSCLC. As compared to the placebo, significant improvement was observed in the ORR and disease control rate (DCR). Furthermore, anlotinib increased the mPFS and OS (PFS 5.37 vs. 1.40 months, OS 9.63 vs. 6.30 months) significantly [32].

According to an exploratory subgroup analysis of the ALTER0303 study, anlotinib significantly improved PFS and OS in patients who have both sensitive wild-type EGFR and EGFR mutations. PFS was 5.57 months vs. 0.83 months (*p* < 0.0001) and OS was 10.70 months vs. 6.27 months (*p* < 0.0227) for patients with sensitive EGFR mutations. Further, PFS was 5.37 months vs. 1.57 months (*p* < 0.0001) and OS was 8.87 months vs. 6.47 months (*p* < 0.0282) for patients with wild-type EGFR [33].

Another clinical study reported that patients with sensitive EGFR mutations improved on anlotinib better than patients with wild-type EGFR (10.70 vs. 8.87, *p* = 0.0204) [34]. Additionally, trials have also demonstrated that anlotinib improves survival rates in older individuals (over 70 years of age) and in patients with squamous cell carcinomas or adenocarcinomas [35,36]. Anlotinib’s benefits for PFS and OS were also unrelated to any prior treatment plans, such as TKIs (gefitinib, erlotinib, and icotinib) or traditional platinum-based chemotherapy [37].

On 8 May 2018, the China Food and Drug Administration (CFDA) approved anlotinib for use as a third-line treatment in advanced NSCLC based on the outcomes of ALTER-0303 [38]. Additionally, for the same indication, anlotinib is advised in the CSCO guidelines for the diagnosis and treatment of primary lung cancer [39].

## 4. Clinical Evidence and Trials of Anlotinib-Based Combination Regimens

Chemotherapy continued to be the mainstay of NSCLC treatment at the start of the twenty-first century. Nonetheless, the advent of immunotherapy has brought out a variety of novel approaches to the management of NSCLC [40,41]. The current standard first-line treatment for NSCLC involves platinum-based chemotherapy, either alone or in combination with immunotherapy [42,43,44]. Immunotherapeutic drugs like pembrolizumab or nivolumab offer effective treatment alternatives for patients who are not responding to chemotherapy as a first-line treatment. On the other hand, patients who were first treated with immunotherapy and chemotherapy are limited to monotherapy alternatives for the following lines of therapy, such as pemetrexed or docetaxel [42,43,44]. Unfortunately, docetaxel-based chemotherapy used as second-line treatment has very unsatisfactory results, with a PFS of just 1.8 to 2.5 months and a median OS of 5.0 to 8.3 months [45,46,47]. So, the combination of chemotherapy with antiangiogenic medicines is a potentially viable option. A new method for treating advanced NSCLC is to combine ramucirumab with docetaxel [48]. However, the effectiveness of the combination of antiangiogenic and chemotherapeutic drugs as a second-line therapy is still doubtful.

There was a real-world study aimed at assessing anlotinib, either alone or in combination, for the management of NSCLC. It included NSCLC patients who received either anlotinib alone or in combination with other drugs retrospectively. There were 240 patients in all groups. The total mPFS was 8.5 (95% CI: 7.1–9.9) months. Compared to anlotinib monotherapy, the combination of immunotherapeutic drugs with anlotinib prolongs the PFS significantly (mPFS: 10.5 vs. 6.5 months). This combination also significantly prolongs the PFS in males who have adenocarcinoma older than 65 years old, with an EGFR wild mutation in stage IV, and metastases outside the thorax. Patients who do not receive antiangiogenesis and have a medical history of hypertension can also benefit from this first-line therapy. Targeted and chemotherapeutic drugs combined with anlotinib had a slightly longer mPFS than anlotinib alone (10.5 vs. 6.5 months, *p* = 0.095, and 9.5 vs. 6.5 months, *p* = 0.177, respectively). Adverse effects were mostly accepted, mild, and could be tolerated, with hypertension being the most frequent [49].

EGFR-TKIs, such as gefitinib, the first or second generation EGFR TKIs, [50,51] and osimertinib, a third-generation EGFR-TKI, [25] have been shown in numerous advanced clinical trials to improve PFS in patients with NSCLC who have an EGFR mutation. For EGFR-mutated NSCLC, EGFR TKIs are now considered the accepted first-line treatment [52]. For advanced EGFR-mutated NSCLC, it is crucial to investigate novel EGFR-TKIs or therapeutic compounds demonstrating biological synergy with EGFR-TKIs, as acquired resistance to these inhibitors invariably develops, and NSCLC patients eventually experience disease progression [53]. Dual inhibition of the VEGF and EGFR signaling pathways offers the possibility of increasing the efficacy of EGFR-targeted therapy and overcoming EGFR-TKI resistance. The VEGF pathway is essential for promoting oncoangiogenesis in lung cancer [54]. While anlotinib inhibits FGFR, PDGFR, c-Kit14, and VEGFR to reduce oncoangiogenesis and tumor growth, bevacizumab only targets the VEGFR signaling pathway [31]. With a low IC_50_ for VEGFR-2 (0.2 nmol/L vs. lenvatinib 4 nmol/L and sorafenib 90 nmol/L) and VEGFR-3 (0.7 nmol/L vs. lenvatinib 5.2 nmol/L and sorafenib 20 nmol/L), anlotinib’s anti-angiogenic properties, when compared to other TKIs, indicate its anticancer activity [28,31,55,56,57]. When anlotinib is used in conjunction with another EGFR-TKI and immune checkpoint inhibitor, sintilimab, anlotinib also shows encouraging anticancer effects in the first-line scenario for advanced NSCLC [58,59]. In a phase 3 study (NCT04028778), 315 patients with treatment-naïve, EGFR-mutated, advanced NSCLC were randomized (1:1) to receive anlotinib or placebo plus gefitinib once daily on days 1–14 per a 3-week cycle. At the prespecified final analysis of PFS, a significant improvement in PFS was observed for the anlotinib arm over the placebo arm (hazards ratio [HR] = 0.64, 95% CI, 0.48–0.80, *p* = 0.003). Particularly, patients with brain metastasis and those harboring EGFR amplification or high tumor mutation load gained significantly more benefits in PFS from gefitinib plus anlotinib. The incidence of grade 3 or higher treatment-emergent adverse events was 49.7% of the patients receiving gefitinib plus anlotinib versus 31.0% of the patients receiving gefitinib plus placebo. Anlotinib plus gefitinib significantly improves PFS in patients with treatment-naïve, EGFR-mutated, advanced NSCLC, with a manageable safety profile [60].

Anlotinib offers a more convenient oral delivery method than intravenous infusion for currently available anti-angiogenic drugs, and it has been studied in the first-line treatment for NSCLC plus chemotherapy [61]. For advanced NSCLC, anlotinib with an EGFR-TKI would be more effective in the first-line setting than EGFR-TKI monotherapy.

NSCLC patients with metastasis or locally advanced disease who were not receiving treatment were recruited for an open-label, three-arm prospective study. NSCLC patients with an EGFR mutation were treated with anlotinib and erlotinib (group 1). Individuals without the EGFR/ALK/ROS1 mutation were given either sintilimab (group 2) or anlotinib with carboplatin plus pemetrexed/gemcitabine (group 3). Safety and ORR were the main results. PFS, DCR, and OS were the secondary objectives. Treatments were administered for a minimum of two cycles, and RECIST version 1.1 was used to assess efficacy. The study’s safety was evaluated at every stage. Grade 3 or higher TRAEs were reported as 77.3% in group 1 and 60% in group 3. The most reported TRAE in group 1 is rash (10.0%), and in group 3, there is a decrease in the number of platelets (30.0%). ORRs were 92.9% in group 1 and 60.0% in group 3, and DCRs were 96.4% in group 1 and 96.7% in group 3. In group 2, the previously published incidence of grade 3 TRAEs or higher and ORR were 54.5% and 72.7%, respectively. For groups 1, 2, and 3, the corresponding median PFSs [95% CI] were 21.6 (15.6 to 24.9), 15.6 (12.9 to NE), and 13.0 [10.5 to not estimated (NE)] months. In group 3, the median OS was 28.1 (95% CI: 21.82 to NE) months. The 24-month OS percentages in groups 1 and 2 were 87.1% and 83.9%, respectively [58]. Future research may confirm that combinations that contain anlotinib with EGFR-TKI, ICI, and chemotherapy are safe and effective as first-line treatments for advanced NSCLC. Patients with advanced NSCLC may have several options for first-line treatment when using a combination based on anlotinib.

A major factor in the morbidity and death of advanced NSCLC is intracranial metastasis that has not responded to standard systemic treatment. Whole-brain radiation (WBRT), a historically developed treatment, is still a highly popular alternative for managing patients with numerous BMs in addition to systemic medicines. Patients who are not candidates for surgery or stereotactic radiosurgery (SRS) can benefit from this intervention. Patients undergoing local therapy with neurosurgical resection or SRS have a restricted number of BMs (≤3) or are only symptomatic [62]. Following WBRT, the disease remission rate ranges from 24% to 55%, and the median OS increases to 3 to 6 months [63]. WBRT is linked to comparatively poor control of pre-existing metastases but has superior distant intracranial tumor control and a significant incidence of late neurocognitive negative effects [62,64,65]. However, there may be no obvious advantage when comparing the WBRT and the supportive treatment alone, due to the poor OS, which occurs with patients of several BMs who are prescribed WBRT. An RCT is examining the WBRT efficacy in comparison with supportive treatment only in NSCLC patients with BMs that are not eligible for SRS. This study shows no great increase in the overall quality of life or OS between the groups [66]. Since WBRT alone continues to provide insufficient therapeutic efficacy in NSCLC patients with BMs who have a poor prognosis, novel treatment approaches are desperately needed.

In patients who usually have symptomatic multiple BMs and are either resistant to previous conventional targeted therapies or lack actionable genetic changes, the combination of WBRT and antiangiogenesis inhibitors may be a helpful approach. In patients with unresectable BMs from solid tumors, the REBECA phase I trial has shown the viability of a combination of conventional WBRT with bevacizumab and has provided preliminary effectiveness data [67]. Anlotinib is very effective in treating intracranial lesions and can help NSCLC patients with BMs who have not responded to at least second-line therapy, according to a post hoc analysis of the ALTER0303 study [68]. These results suggest that WBRT plus anlotinib might be a worthwhile approach for treating BMs from NSCLC and call for more research. Despite the successful effect of anlotinib in certain advanced lung cancer cases with BM [68], cranial radiotherapy (CRT) is currently regarded as the standard treatment regimen for NSCLC patients without a specific gene mutation or EGFR/ALK/ROS1-TKI resistance. It is due to its effect in expeditiously relieving the central nervous system symptoms and lengthening the patients’ survival period [69]. CRT can increase the blood–brain barrier’s (BBB) permeability, which may increase the brain tissue’s amount of anlotinib [70]. For NSCLC patients without a specific gene mutation or EGFR/ALK/ROS1-TKI resistance, the curative effect of CRT plus anlotinib may be superior to that of CRT alone.

## 5. Anlotinib and Chemotherapy Combinations for NSCLC

There are several combinations of anlotinib with chemotherapy that were assessed for the treatment of NSCLC. Some of these combinations include anlotinib with chemotherapeutic drugs that contain platinum (cisplatin or carboplatin) and pemetrexed, anlotinib with docetaxel, and anlotinib with S-1.

First- or second-generation EGFR TKIs were ineffective in treating patients with T790M-negative EGFR-mutant advanced non-squamous NSCLC. Therefore, a phase 1b/2 trial was conducted in multiple centers to determine the activity, safety, and maximum tolerated dose (MTD) of anlotinib and the chemotherapy combination. For a three-week cycle based on a three + three dose-escalation design, anlotinib was given (8/10/12 mg, days 1–14) in combination with either carboplatin (AUC = 5, day 1) or cisplatin (75 mg/m^2^, day 1) with pemetrexed (500 mg/m^2^, day 1). Anlotinib maintenance therapy was given after four cycles of anlotinib with chemotherapy plus pemetrexed at MTD in the single-armed phase 2 trial. PFS was the main goal of the phase 2 stage. Slow enrolment led to the early termination of the study after only 19 patients were enrolled. The mPFS was 5.75 (95% CI: 4.37–7.52) months. Anlotinib’s MTD was 12 mg. The ORR was 47% (95% CI: 24.5–71.1%). Seven patients (58.3%) in the 12 mg group suffered grade 3–4 TRAEs, with reduced platelet count, hypertriglyceridemia, and hypertension being the most common. There were no treatment-related fatalities [71].

To investigate the safety and effectiveness of anlotinib in combination with chemotherapy that contains carboplatin/ pemetrexed, followed by maintenance therapy (anlotinib + pemetrexed), a clinical trial was carried out in advanced EGFR/ALK wild-type nsq-NSCLC. In Henan Province, patients with advanced NSCLC of wild-type EGFR/ALK were enrolled. Every patient received maintenance therapy (anlotinib plus pemetrexed) after receiving anlotinib in conjunction with carboplatin/pemetrexed-based chemotherapy. PFS was the main outcome. ORR, OS, DCR, and AEs were the secondary objectives. The survival follow-up was conducted every six weeks. Thirty-eight participants, ranging in age from 33 to 75 years, were assessed, with a median age of 62. Finally, five patients were still receiving maintenance therapy. The median OS was 23.4 months, while the mPFS was 10.5 (95% CI: 4.1, 17.0) months. There was a 94.7% DCR and an ORR of 60.5%. Twelve participants experienced TRAEs of grade three or higher. Hypertension (23.7%), neutropenia (19.4%), and bone marrow toxicity (10.5%) were the most frequent TRAEs. Out of the seven patients, two stopped their treatment during induction, and the other five throughout maintenance. There were no recorded grade 5 TRAEs [72].

Oral S-1 is a third-generation fluorouracil derivative that has good efficacy and comparatively minimal toxicity when treating stage IV NSCLC. Simon developed a phase II clinical trial to assess the safety and effectiveness of anlotinib plus S-1 as a third treatment for patients with stage IV NSCLC. The clinical trial had 29 participants in total. Over 21 days, anlotinib was continued until the disease progressed or the patient passed away if the efficacy was determined until there was a complete or partial response after six cycles or a stable disease. The ORR served as the main outcome. Of the 29 patients, 30% achieved the intended primary endpoint of ORR. The mPFS and OS durations were 5.8 and 16.7 months, respectively. AEs, such as diarrhea, nausea, and vomiting, as well as fatigue and hypertension, were the most frequent. There were no AEs in grade 4 or treatment-related deaths [73]. Thus, anlotinib and S-1, the third therapy for stage IV NSCLC, exhibit tolerable toxicity and promising antitumor activity in patients with NSCLC.

Patients with previously treated NSCLC underwent a randomized trial between anlotinib + docetaxel vs. docetaxel. PFS was the primary endpoint, while the secondary objectives were safety, OS, ORR, and DCR. The combination of anlotinib and docetaxel significantly increased median progression-free survival (mPFS) (95% CI: 0.23–0.63, *p* = 0.0002), and showed better DCR (87.5% vs. 53.5%, *p* = 0.0007) and ORR (32.5% vs. 9.3%, *p* = 0.0089). The combination group’s median OS was 12.0 months (*p* = 0.4803).

Patients who had previously received immunotherapy had a significant mPFS (7.8 months vs. 1.7 months, *p* = 0.0290). Both groups experienced a manageable incidence of grade ≥ 3 TRAEs, primarily neutropenia (10.0% vs. 5.0%) and leukopenia (15.0% vs. 7.0%) [74]. This means that for advanced NSCLC, which did not respond to first-line platinum-based therapy, anlotinib combined with docetaxel presents a promising new treatment option.

All of these studies point to anlotinib as a potentially effective therapeutic option for patients with advanced NSCLC, especially if they have not responded to first- or second-line EGFR TKIs or have certain genetic profiles. Anlotinib’s effectiveness in these situations, combined with a tolerable safety profile, highlights the drug’s potential as a flexible part of NSCLC therapy plans. Future studies and clinical trials will clarify the function of anlotinib and maximize its application in the context of NSCLC treatment.

## 6. Anlotinib and Immunotherapy Combinations for NSCLC

Antiangiogenic inhibitors and immunotherapeutic medicines together have shown synergistic antitumor effects in several cancer types, opening the door to the investigation of immunotherapy in combination with antiangiogenic therapies for advanced tumors [75].

The development of immune checkpoint inhibitors (ICI) has significantly altered the therapy landscape for NSCLC over the past ten years. Co-stimulating surface proteins on T cells that deliver inhibitory signals are known as immune checkpoints. The immune system can be evaded by tumor cells by interfering with immunological checkpoints, which delays T cell activation and the lethal effects of T cells on tumors [76]. Antibodies can readily block immunological checkpoints since they are cell surface receptor proteins. The most effective of these antibodies are anti-programmed death-1 (PD-1)/programmed death-ligand 1 (PD-L1) antibodies, which are authorized to treat a broad range of malignancies, including those of the blood, skin, lung, liver, bladder, and kidney [77,78].

China uses a domestic, totally human IgG4 monoclonal antibody called sintilimab to fight PD-1. The National Medical Products Administration (NMPA) first approved it for the treatment of patients with classical Hodgkin’s lymphoma who were refractory or had relapsed after receiving at least two lines of systemic chemotherapy [79]. Later on, the NMPA approved the use of sintilimab as a first-line treatment for hepatocellular carcinoma when combined with IBI305 and for NSCLC in conjunction with chemotherapy [80]. In addition, regimens containing sintilimab as first-line therapies were approved for the treatment of NSCLC because of findings from significant phase 3 NSCLC studies (ORIENT-11 and ORIENT-12) [81,82].

Patients who experienced stage IV, previously untreated, EGFR/ALK/ROS1 negative NSCLC were recruited in a phase II study in six centers. Patients were classified by PD-L1 expression and histology and randomly assigned to group A (sintilimab 200 mg day 1, anlotinib 12 mg, QD 1–14) or group B (platinum-based chemotherapy; cross-over to sintilimab was allowed after disease progression [PD]). Every three weeks, treatment was administered until PD, extreme toxicity, withdrawal, or passing away. One could receive sintilimab for up to 24 months or 35 cycles. ORR was the main goal; PFS, DCR, duration of response (DoR), OS, and safety were the secondary outcomes. A total of 89 patients were randomized between November 2019 and July 2022 (43 to group A, 46 to group B). On July 15, 2022, 84 patients were evaluable (41 vs. 43), with a median follow-up of 13.1 months. With a DoR of 16.3 vs. 6.2 mo, group A’s ORR was 50.0% (95% CI [33.8–66.2]) while group B’s was 32.6% (95% CI [19.1–48.5]). The mPFS was 10.8 vs. 5.7 mo (95% CI [0.25–0.74]), and the DCR was 85.0% (95% CI [70.2–94.3]) vs. 93.0% (95% CI [80.9–98.5]). OS lacked maturity. In group A, 11.6% of patients experienced grade 3–4 TRAEs, compared to 43.5% in group B. In group A, elevated AST, hyponatremia, and hypothyroidism were the most common findings. Two patients had their treatment stopped, and one patient died because of the development of TRAEs [83].

Further investigation revealed a clinically significant relationship between the therapeutic results of anlotinib with PD-1 blockades and the degree of response to prior immunotherapeutic drugs. A retrospective exploratory trial examined anlotinib in combination with PD-1 blockades for advanced NSCLC in patients who were previously treated with immunotherapeutic drugs. Sixty-seven patients with advanced NSCLC were treated with anlotinib and PD-1 blockades in a clinical setting. The four PD-1 blockades employed in this study (pembrolizumab, camrelizumab, sintilimab, and tislelizumab) were all licensed in China [84]. Out of the 67 patients, the best overall response indicated that 10 patients had progressive disease, 16 patients had a partial response, and 41 had stable disease. This resulted in an ORR of 23.9% (95% CI: 14.3–35.9%) and a DCR of 85.1% (95% CI: 74.3–92.6%). The prognostic results showed a median OS of 16.5 months (95% CI: 10.73–22.27) and a PFS of 6.1 months (95% CI: 2.37–9.83). According to an exploratory analysis, 17 patients who could not previously tolerate the immunotherapeutic treatment may have had a better prognosis (median OS: 22.3 months vs. 12.5 months, *p* = 0.024). Furthermore, 62 patients (92.5%) experienced adverse responses of any grade when receiving anlotinib plus PD-1 blockades; of them, 31 patients (46.3%) experienced adverse reactions of ≥ grade 3. The most common AEs were hepatotoxicity, tiredness, diarrhea, and hypertension. Individuals with advanced NSCLC who had received prior immunotherapy showed promising efficacy and manageable tolerability with anlotinib with PD-1 blockades. Patients who did not respond well to prior immune-related regimens may benefit greatly from anlotinib plus PD-1 blockades. Future research should verify this conclusion.

The results of the previous two studies show that anlotinib with sintilimab offers a promising therapeutic alternative for patients with advanced NSCLC who have not received treatment previously. This combination shows greater efficacy and a more favorable safety profile. Anlotinib plus PD-1 blockades may also be beneficial for patients with NSCLC who have tried immunotherapy in the past, especially if the patients did not respond well to earlier immune-related regimens. These results bolster the need for additional investigation to validate these findings and may broaden the range of therapeutic alternatives available to NSCLC patients.

The safety and effectiveness of anlotinib-containing regimens from the published clinical trials are summarized in Table 1.

## 7. Addressing Challenges That Hinder the Use of Anlotinib

Advanced NSCLC treatment is a challenge due to drug resistance. The prognosis and management of advanced NSCLC can be improved by knowing the exact mechanisms of drug resistance, such as anlotinib or osimertinib. The therapeutic effectiveness of patients with EGFR-activating mutations is significantly restricted by the inevitable development of EGFR-TKI resistance [88,89].

Despite its clinical efficacy, resistance to anlotinib can emerge through various mechanisms. Tumor cells may bypass VEGFR inhibition by upregulating compensatory pro-angiogenic pathways such as the FGF/FGFR axis or hepatocyte growth factor (HGF)/c-MET signaling [31,90]. Acquired mutations or amplification in receptor tyrosine kinases (e.g., FGFR, PDGFR, or c-Kit) can reduce anlotinib’s inhibitory effect, contributing to resistance [91]. Hypoxia induced by antiangiogenic therapy can lead to adaptive changes in the tumor microenvironment, promoting resistance through enhanced epithelial–mesenchymal transition (EMT), immune evasion, or stromal activation [92,93]. Overexpression of ATP-binding cassette (ABC) transporters (e.g., P-gp) may reduce intracellular drug accumulation, decreasing anlotinib efficacy [94].

Therefore, promising clinical data points to combination therapy as the upcoming wave of novel treatments for advanced NSCLC. There is a preclinical study to assess the anticancer effects of anlotinib and gefitinib in vivo and in vitro models of gefitinib-resistant lung adenocarcinoma. They examined the therapeutic impact of anlotinib and EGFR-TKI therapy in 24 patients with advanced EGFR-mutant NSCLC. By boosting gefitinib’s pro-apoptotic and anti-proliferative properties, anlotinib was able to reverse gefitinib resistance in lung cancer animals that were resistant to the drug. So, the combination therapy of gefitinib and anlotinib exhibited a synergistic anti-tumor impact. Anlotinib plus EGFR-TKI therapy had a DCR of 95.8% and an ORR of 20.8%. Although the median OS was not reached, the mPFS was 11.53  ±  2.41 months [95].

Twenty patients with advanced NSCLC who had developed possible acquired drug resistance after receiving gefitinib or icotinib were enrolled in a clinical trial that was conducted from April 2018 to June 2020. The targeted medicine was combined with anlotinib at the initial dosage. Every eight weeks, patients had CT scans, and the associated AEs and curative outcomes were noted. Eight patients were in a stable state and were receiving treatment after anlotinib was administered; the best efficacy DCR was 100%. The mPFS was 15.7 months (10.19–18.87 months). Seven patients have shown a significant decrease in the tumor marker carcinoembryonic antigen levels. Following the administration of anlotinib, the most common AEs that were observed included oral ulcers, tiredness, diarrhea, hand-foot-skin reactions, hypertension, and anorexia. So, in vivo, tumor angiogenesis was reduced by the combination of gefitinib and anlotinib with just a few AEs. It also markedly increased therapy efficacy for some patients, delaying PD and enhancing survival. For patients with advanced NSCLC who are resistant to EGFR-TKI and may also be resistant to subsequent drugs, this medication combination appears hopeful [96].

According to anlotinib resistance, researchers have discovered that a poor anlotinib response is linked to elevated plasma exosomal miR-136-5p levels in NSCLC patients. NSCLC cells that resist anlotinib can transfer functional miR-136-5p to parent NSCLC cells via exosomes, promoting the parent cells’ proliferation. Targeting PPP2R2A, exosomal miR-136-5p activates the Akt pathway and confers anlotinib resistance [97]. This mechanism suggests that miR-136-5p could serve as a possible biomarker for anlotinib response in NSCLC.

Another challenge hindering the involvement of anlotinib in the treatment regimens of NSCLC is anlotinib toxicity. In the phase I trial, every adverse event (AE) was under control. Hand–foot skin reaction (HFSR) had the highest incidence of adverse events (AEs) with 53%; other common AEs included proteinuria (67%), triglyceride elevation (62%), ALT elevation (48%), serum amylase (43%), AST elevation (43%), total bilirubin elevation (38%), hypertension (34%), and leukopenia (33%) [98]. Anlotinib’s overall rate of AEs was 100%. Of the patients who experienced AEs, 29% had grade 3 out of 4 reactions, which included hypertension (10%), triglyceride elevation (10%), lipase rise (5%), and hand–foot skin reaction (5%) [98].

Hypertension is among the most common AEs due to VEGFR inhibition, occurring in 40–60% of patients [27]. Moreover, HFSR is a frequent dermatologic toxicity seen in >30% of patients [98]. Proteinuria occurs in up to 20–30% of cases due to glomerular endothelial dysfunction [99]. Hyperlipidemia (especially cholesterol elevation) is reported in over 40% of patients, likely due to metabolic modulation by VEGF/FGFR inhibition [99]. Fatigue, anorexia, and diarrhea are non-specific but commonly observed AEs. Even if venous or arterial thromboembolism has been reported to be less frequent, it still has to be closely monitored [100].

Notably, compared to other oral anti-VEGFR TKIs, anlotinib seems to induce less and milder diarrhea, as reported [101,102,103]. It should be mentioned that triglyceride and cholesterol elevations were common in anlotinib-treated patients. The authors recommended that patients taking anlotinib should undergo routine monitoring, even though these side effects did not cause any noticeable symptoms. This is especially important since some AEs are linked to thromboembolism, which is more common with anti-VEGFR TKI drugs [104]. Patients with refractory advanced NSCLC taking anlotinib as a third-line treatment experienced grade 3 or higher adverse events (AEs) in a phase III clinical trial. However, no treatment-related deaths were reported [32].

## 8. Future Perspectives

Exceptional efficacy and manageable toxicity characterize anlotinib’s use in the treatment of advanced NSCLC. On 8 May 2018, anlotinib was initially approved in China for patients with advanced NSCLC who had not improved following at least two lines of previous treatment.

Even though anlotinib has demonstrated efficacy against NSCLC, several issues still need to be resolved before the drug is widely used. Initially, more research on predictive indicators is necessary to help choose the best candidates for anlotinib therapy. There are not enough predictive biomarkers overall, even though several biomarkers seem to identify patients who will most likely benefit from anlotinib treatment. Moreover, one clinical study that employed a small number of samples from NSCLC patients revealed the present possible biomarkers. There are still no reliable biomarkers for other forms of cancer. Subsequent studies must delve deeper into determining the best indications for those who stand to gain from anlotinib.

Secondly, further research is required to determine the ideal dosing schedule for anlotinib in conjunction with other therapies. Further research is necessary to determine whether anlotinib can be used as a first-line treatment for NSCLC or not. Anlotinib is a novel drug, and as such, its efficacy is thought to be limited. Moreover, anlotinib is anticipated to demonstrate effectiveness against further tumor types. As a result, further excellent randomized trials ought to be carried out to determine its therapeutic efficacy in other illnesses. Furthermore, anlotinib may have varying therapeutic effects on various cancer types. Therefore, more research is required to determine the best anlotinib treatment plan for these tumors.

Finally, there is still much to learn about the long-term toxicity profile of anlotinib. Therefore, it is important to clarify any potential long-term harm as anlotinib trials increase in number. Because anlotinib studies have only recently begun, we know very little about tumor resistance to anlotinib and its potential causes. On the other hand, understanding how to identify and treat anlotinib resistance is crucial. So, subsequent studies are needed to devise tailored treatment approaches to overcome resistance.

Table 2 presents the most recent protocols of the trials that used the anlotinib either as monotherapy or in a combination regimen.

## 9. Conclusions

This comprehensive review highlights the growing evidence supporting the effectiveness of anlotinib-based therapy regimens for NSCLC, particularly when combined with other therapeutic modalities. Notable combination strategies, such as anlotinib with platinum-based chemotherapy and anlotinib with PD-1 inhibitors, have demonstrated significant improvements in overall survival, progression-free survival, and objective response rate. However, the treatment of NSCLC is hindered by challenges such as drug resistance and the toxicity profile of therapies. Although anlotinib’s toxicity profile is generally manageable, adverse events such as hand-foot skin reactions, proteinuria, and cholesterol elevation necessitate regular monitoring, particularly due to the risk of thromboembolic events associated with anti-VEGFR TKIs. These challenges highlight the need for ongoing research to refine treatment strategies. Ultimately, incorporating anlotinib into multimodal treatment regimens offers the potential to significantly improve patient outcomes in NSCLC, paving the way for personalized and targeted management of this complex disease.

## Figures and Tables

**Figure 1 pharmaceuticals-18-00585-f001:**
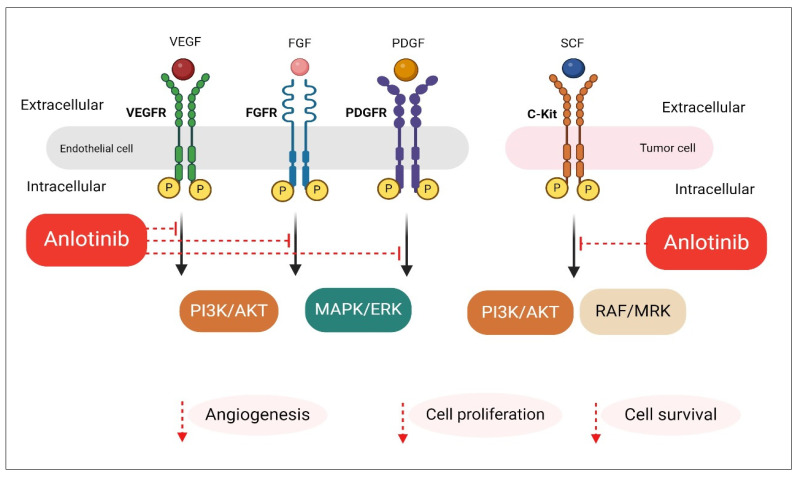
Mechanism of action of anlotinib. The newly developed oral small-molecule receptor tyrosine kinase inhibitor anlotinib targets VEGFR-1, VEGFR-2/KDR, VEGFR-3, c-Kit, PDGFR-α, and the fibroblast growth factor receptors (FGFR1, FGFR2, and FGFR3). It can also suppress the growth of tumor cells and angiogenesis. Created with www.BioRender.com.

**Table 1 pharmaceuticals-18-00585-t001:** Summary of clinical trials that assess the efficacy and safety of anlotinib-based combinations.

Types of Combination	Intervention	Study ID	Type of the Study	Number of Patients	Efficacy					Safety TRAEs (%)
					mPFS (Month) (95 % CI)	OS (Months)	OSR (Percentage)	ORR (%)	DCR (%)	
Anlotinib and chemotherapy combinations	Anlotinib with platinum-based Chemotherapy and pemetrexed	Li et al. 2022 [71]	A multicenter phase 1b/2 trial	19	5.75 (4.37–7.52)	NA	47.4%	NA *	NA	Hypertension (50.0%). Decreased platelet count (16.7%). Hypertriglyceridemia (8.3%).
	Anlotinib and carboplatin combined with pemetrexed	He et al. 2022 [72]	A multicenter, single-arm trial	38	10.5 (4.1–17.0)	NA	23.4%	60.5%	94.7%	Hypertension (23.7%). Neutropenia (19.4%). Bone marrow toxicity (10.5%).
	Anlotinib Combined with S-1	Xiang et al. 2021 [73]	A Phase II Clinical Trial	29	5.8	16.7	NA	30%	NA	Fatigue (55%) Hypertension (38%) Liver dysfunction/failure (clinical) (34%) Hypothyroidism (31%)
	Anlotinib plus docetaxel	Pu et al. 2024 [74]	A multicenter, randomized phase II trial	40	4.4	12	NA	32.5%	87.5%	TRAEs (82.5%) include Fatigue Anemia Leukopenia Grade ≥3 TRAEs occur in 30.0 %.
Anlotinib + Immunochemotherapy	Sintilimab and anlotinib	Han et al. 2022 [83]	An open-label, multicenter, randomized, phase II study	41	10.8 (0.25–0.74)	NA	NA	50%	85%	Grade 3-4 TRAEs (11.6 %) include: Hypothyroidism Hyponatremia AST elevation.
	Anlotininb combined with PD-1 blockades (sintilimab, ocrelizumab, tislelizumab, and pembrolizumab)	Dou et al. 2024 [84]	A Retrospective Exploratory Study	67	6 (2.37–9.83)	16.5	NA	23.9%	85.1%	
	EGFR-TKI + Anti-angiogenesis (Gefitinib + Anlotinib)	Zhou et al. 2024 [60]	Multicenter, double-blind, randomized Phase 3 trial	157	14.8 (12.9–15.4)	31.2 (25.7–NE)	NA	76.1 (68.6–82.6)	NA	Grade ≥ 3 (49.7%) Hypertension (29.7%) Diarrhea (66.5%) Rash (65.8%)
Anlotinib + Immunochemotherapy	Anlotinib (antiangiogenic TKI) + EGFR-TKIs (1st/2nd/3rd generation)	Chen et al. 2025 [85]	Open-label, single-arm, multicenter, phase II trial.	120	9.1 (6.8–11.7)	81.1 (71.8–87.5)	NA	6.7	87.5	All-grade: 96.7% (116/120). Grade ≥ 3: 52.5% (63/120). Most common: Hypertension (19.2%) Diarrhea (5.0%) Weight loss (4.2%). Discontinuations due to AEs: 12.5% (15/120). Serious AEs: Hemoptysis (4 cases), interstitial lung disease (1 case).
	Perioperative immunotherapy (sintilimab) + neoadjuvant antiangiogenic therapy (anlotinib) + chemotherapy (platinum-based doublet).	Duan et al. 2024 [86]	Open-label, single-arm, phase 2 trial.	45	81.5 (64.5–90.9) (24 months).	97.7 (84.6–99.7) (12 months)	NA	71.1	97.8	All-grade: 100% (45/45). Grade 3/4: 55.6% (25/45). Most common: White blood cell decrease (11.1%). Neutrophil count decrease (11.1%). Vomiting (8.9%). Immune-Related AEs (irAEs): Neoadjuvant phase: 15.6% (7/45). Adjuvant phase: 34.1% (14/41), with grade 3 in 17.1%.
	immunotherapy (sintilimab), antiangiogenic therapy (anlotinib), and chemotherapy (nab-paclitaxel)	Han et al. 2024 [87]	Open-label, single-arm, phase II clinical trial.	25	6.0 (5.4–9.7)	62.2 (12 months)	NA	60	76	All-grade: 92% (23/25). Most common: Leukopenia (56%), Anemia (52%) Elevated GGT (48%). Grade ≥ 3: 16% (4/25). Elevated AST (12%) Rash (4%). Serious AEs: 12% (3/25). Discontinuations due to AEs: 8% (2/25). irAEs: 44% (11/25), most frequently hypertriglyceridemia (16%).

mPFS: Median progression-free survival, OS: overall survival presented in months, OSR: overall survival rate presented in percentage, ORR: objective response rate, DCR: disease control rate, and TRAEs: treatment-related adverse event. * NA: not applicable, NE: Not Estimable.

**Table 2 pharmaceuticals-18-00585-t002:** Summary of the most recent clinical trials about the use of anlotinib in NSCLC.

Protocol ID	Clinical Phase	Study Status	Combination Partners	Key Endpoints
NCT02388919	Phase 2 Phase 3	Completed	Anlotinib used as monotherapy.	Evaluate the efficacy and safety of anlotinib as the 3-line treatment of patients with advanced non-small lung cancer, with placebo control.
NCT04967079	Phase 1	Completed	MEK inhibitor trametinib (2 mg) in combination with anlotinib (6 mg, 8 mg, 10 mg, 12 mg).	In part A, the primary endpoint is the determination of the recommended RP2D. Secondary endpoint for phase Ia includes evaluating the ORR, DCR, PFS, and AEs. Following the establishment of the RP2D, the expansion cohort will be initiated. Transitioning to part B, 20 patients will be enrolled to further evaluate the ORR.
NCT06188650	NA	Recruiting	DEB-BACE combined with anlotinib and adebelimumab.	The goal of this clinical trial is to learn about DEB-BACE combined with anlotinib and adalimumab in patients with advanced NSCLC after second-line treatment.
NCT03765775	Phase 2	Unknown	Anlotinib plus sintilimab.	This is an efficacy and safety study of anlotinib combined with sintilimab (IBI 308) in participants with advanced or metastatic NSCLC who have resistance against first-generation EGFR-TKIs, along with T790M negative.
NCT04211896	Phase 2	Unknown	Anlotinib combined with nivolumab.	This study evaluates the safety and efficacy of anlotinib in combination with nivilumab as a second-line treatment in advanced NSCLC patients. The primary endpoint of the study is PFS; the secondary endpoints are DCR, ORR, OS, and safety.
NCT05460481	Phase 2	Unknown	Anlotinib plus penpulimab.	The investigation of the efficacy and safety of anlotinib plus docetaxel in advanced NSCLC patients who have progressed following prior PD-1 or PD-L1 inhibitor treatment.
NCT01924195	Phase 2	Completed	Anlotinib used as monotherapy.	The trial is to explore anlotinib for the effectiveness and safety of advanced non-small cell lung cancer patients who have failed two lines of chemotherapy.

Abbreviations: ORR: objective response rate, RP2D: phase 2 dose, DCR: disease control rate, PFS: progression-free survival, AEs: adverse events, DEB-BACE: drug-loaded microspheres for bronchial artery chemoembolization, NSCLC: non-small cell lung cancer, OS: overall survival, EGFR-TKIs: epidermal growth factor receptor tyrosine kinase inhibitors, PD-1: programmed death-1 protein, NA: Not Applicable.

## Data Availability

Not applicable.

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
