# Peer review of "Synergizing Success: The Role of Anlotinib Combinations in Advanced Non-Small Cell Lung Cancer Treatment"

_pharmaceuticals, 2025, doi:10.3390/ph18040585_

Round 1
Reviewer 1 Report
Comments and Suggestions for Authors
Hetta et al. systematically reviewed recent clinical trial advances in anlotinib combinations in NSCLC in the manuscript. However, several improvements are still needed before publication. Here are my comments:
- The authors only discussed trials up to April 2024, more recent clinical trial reports should be included.
- Although the authors have summarized combination therapies with chemotherapy and immunotherapy in Table 1, studies on combinations with EGFR-TKIs should also be added.
- The authors should also include data from clinicaltrials.gov and summarize anlotinib trials (both monotherapy and combination) with details such as id, clinical phase, study status, combination partners, and key endpoints.
- The authors should also discuss combination strategies other than chemotherapy, immunotherapy, and EGFR-TKIs, such as radiotherapy.
- Please correct the error in reference 59.
Author Response
Point-by-point response
Reviewer 1:
- Comment: The authors only discussed trials up to April 2024; more recent clinical trial reports should be included.
Response
- Thank you for your comment. We appreciate your suggestion and acknowledge the importance of including the most recent clinical trial reports. We updated our review to incorporate relevant trials published up to 28 March 2025 to ensure our analysis remains current.
- We added the most recent trials in Table 1.
- Comment: Although the authors have summarized combination therapies with chemotherapy and immunotherapy in Table 1, studies on combinations with EGFR-TKIs should also be added.
Response
- Thank you for your comment. We have now included the more recent clinical trial, “Gefitinib (an EGFR tyrosine kinase inhibitor) plus anlotinib (a multikinase inhibitor) for untreated, EGFR-mutated, advanced non-small cell lung cancer (FL-ALTER): a multicenter phase III trial”, in Table 1 that summarizes the clinical trial updates. This addition ensures our review is up to date with the latest advancements.
- Additionally, we discussed the combination of anlotinib and EGFR-TKIs in section “5. Clinical Evidence and Trials of Anlotinib-Based Combination Regimens” Lines: 224-253.
- Comment: The authors should also include data from clinicaltrials.gov and summarize anlotinib trials (both monotherapy and combination) with details such as ID, clinical phase, study status, combination partners, and key endpoints.
Response
- Thank you for your valuable comment. In response, we have now included the recent anlotinib trials registered on ClinicalTrials.gov. This includes both monotherapy and combination therapy studies. For each trial, we provide the ClinicalTrials.gov identifier, clinical phase, current study status, combination partners (if applicable), and key study endpoints. This information has been incorporated into the revised manuscript in Table 2.
- Comment: The authors should also discuss combination strategies other than chemotherapy, immunotherapy, and EGFR-TKIs, such as radiotherapy.
Response
- Thank you for this insightful suggestion. In the revised manuscript, we have expanded the discussion to include combination strategies beyond chemotherapy, immunotherapy, and EGFR-TKIs. Specifically, we now address the potential of combining anlotinib with radiotherapy in patients with brain metastasis in section 5. Clinical Evidence and Trials of Anlotinib-Based Combination Regimens, Lines 296-331. This addition provides a more comprehensive overview of emerging therapeutic strategies and aligns with the evolving landscape of multimodal cancer treatment.
- Comment: Please correct the error in reference 59.
Response
- Thank you for pointing this out. We have reviewed and corrected the error in reference 59 to ensure accuracy and consistency with the citation format used throughout the manuscript.

Reviewer 2 Report
Comments and Suggestions for Authors
This is a focused review on Anlotinib in advanced non-small cell lung cancer (NSCLC) treatments. I have a few suggestions:
- The authors can include information on the prevalence and geographical distribution of NSCLC in the introduction section.
- Please check reference no. 59.
Author Response
Reviewer 2:
- Comment: The authors can include information on the prevalence and geographical distribution of NSCLC in the introduction section.
Response:
- Thank you for your comment. In response, we have revised the introduction to include information on the global prevalence and geographical distribution of non-small cell lung cancer (NSCLC). This addition provides important context regarding the burden of disease across different regions and highlights the relevance of continued therapeutic development, including treatments such as anlotinib. Introduction section, Lines 57-72.
- Comment: Please check reference no. 59
Response
- Thank you for pointing this out. We have reviewed and corrected the error in reference 59 to ensure accuracy and consistency with the citation format used throughout the manuscript

Reviewer 3 Report
Comments and Suggestions for Authors
General Comments
Despite having an interesting topic, the draft fails to elaborate on its aims. The English language of the draft is incomprehensible at some points, and the body of the draft is unorganized. Most parts have been added without logical narration, and no robust conclusions are drawn. In addition, the draft does not cover the recent publications in the field, which is accompanied by its flawed search strategy. Therefore, according to my assessment, the manuscript is not qualified for further consideration.
Specific Comments
Major Issues
- p. 2, ll. 64-71: Most of these explanations are irrelevant to the topic of the manuscript and should be removed.
- p. 2, ll. 82-93: Therapeutic options for cases with advanced/refractory NSCLC should be discussed more thoroughly.
- The search strategy provided in the Methods section is wrongly crafted, as there are redundancies (e.g., with Lung Carcinomas, there is no need for Non-Small-Cell Lung Carcinomas), and terms like “Novel Targeted Drug” are not appropriate. Added to this, the search strategy for each database is not provided. Nonetheless, given the fact that the manuscript is not a systematic review, having a Methods section is deemed unnecessary in the first place.
- Section 3 gives a shallow overview of anlotinib’s mechanisms of action. This section must be elaborated.
- As mentioned above, the draft has insufficient depth in discussing key literature. In fact, the authors have picked a limited number of trials and just summarized their objective outcomes, without interpreting them. Such numbers should be put in tables, and lucid explanations and conclusions must be made.
- Likewise, the mechanisms of resistance and side effects of anlotinib are not discussed enough.
Minor Issues
- Each abbreviation must be defined by its first appearance within the draft. This has been violated several times (e.g., p. 2, l. 60, and p. 2, l. 95).
Some parts of the draft need polishing to improve the readability. For example, the following sentence can be shortened:
"Although there is a difference between the two groups in the OS, it is an insignificant difference (P = 0.2316)."
The following sentence, as another example, is hard to read (i.e., chemotherapy that contains):
"The approved first-line treatment paradigm for NSCLC is now chemotherapy that contains platinum, either in isolation or in conjunction with immunotherapeutic drugs."
Author Response
Reviewer 3:
- Comment: Despite having an interesting topic, the draft fails to elaborate on its aims.
Response
- Thank you for your response. In the revised manuscript, we have clearly stated the aim of the study in the introduction section, lines 118-122. Specifically, we now emphasize that the objective is to comprehensively review and evaluate the role of anlotinib in advanced non-small cell lung cancer (NSCLC), both as a monotherapy and in combination with other treatment strategies such as chemotherapy, radiotherapy, EGFR-TKIs, and immunotherapy. This clarification helps set the scope and direction of the review more effectively.
- Comment: The English language used in the draft is sometimes incomprehensible, and the body of the draft is disorganized.
Response
- We sincerely appreciate your feedback regarding the clarity and organization of the manuscript. In response, we have thoroughly revised the language to improve readability and ensure that the content is presented in a more coherent and organized manner. This includes refining sentence structures, simplifying complex expressions, and ensuring smooth transitions between sections.
- Comment: Most parts have been added without logical narration, and no robust conclusions are drawn.
Response
- We appreciate the reviewer’s constructive feedback. In response, we have reorganized the manuscript to ensure a more logical flow of information. We have made efforts to strengthen the narrative by clearly linking sections and providing smoother transitions between topics. Additionally, we have revised the conclusion to more effectively summarize the key findings and implications of the study, ensuring that robust and meaningful conclusions are drawn based on the reviewed evidence. Conclusion section, Lines 571-585.
- Comment: In addition, the draft does not cover the recent publications in the field, which is accompanied by its flawed search strategy.
Response
- Thank you for your comment. We appreciate your suggestion and acknowledge the importance of including the most recent clinical trial reports. We updated our review to incorporate relevant trials published up to 28 March 2025 to ensure our analysis remains current.
- We added the most recent trials in Table 1.
- Comment: p. 2, ll. 64-71: Most of these explanations are irrelevant to the topic of the manuscript and should be removed.
Response
- Thank you for your valuable feedback. Upon review, we have removed the sections on pages 2, lines 64-71, as they were not directly relevant to the focus of the manuscript. This revision has helped streamline the content and ensure that the manuscript remains focused on the primary topic.
- Comment: p. 2, ll. 82-93: Therapeutic options for cases with advanced/refractory NSCLC should be discussed more thoroughly.
Response
- Thank you for this important observation. In response, we have revised the paragraph to provide a more focused and thorough discussion of therapeutic options for advanced and refractory NSCLC. The updated version emphasizes current standard treatments, targeted therapies, and novel approaches relevant to this disease stage, aligning more closely with the scope of the manuscript. Introduction section, lines
- Comment: The search strategy provided in the Methods section is wrongly crafted, as there are redundancies (e.g., with Lung Carcinomas, there is no need for Non-Small-Cell Lung Carcinomas), and terms like “Novel Targeted Drug” are not appropriate. Added to this, the search strategy for each database is not provided. Nonetheless, given the fact that the manuscript is not a systematic review, having a Methods section is deemed unnecessary in the first place.
Response
- Thank you for this valuable observation. We acknowledge the redundancy and imprecision in the original search strategy, including the use of overlapping and inappropriate terms. In light of your comment and considering that this manuscript is a narrative review rather than a systematic review, we have removed the Methods section entirely. Instead, relevant details regarding the literature scope and selection have been briefly summarized in the Introduction to maintain transparency while aligning with the expectations for a narrative review format.
- Comment: Section 3 gives a shallow overview of anlotinib’s mechanisms of action. This section must be elaborated.
Response
- Thank you for your insightful comment. In response, we have substantially revised and expanded Section 3, which now becomes Section 2 after deleting the Method section, to provide a more comprehensive and in-depth overview of anlotinib’s mechanisms of action. The revised section now includes detailed descriptions of the key signaling pathways targeted by anlotinib, such as VEGFR, FGFR, PDGFR, and c-Kit, and explains how their inhibition contributes to antiangiogenic, antiproliferative, and antimetastatic effects. We have also included relevant molecular insights and updated references to better support the mechanistic explanation. These additions aim to enhance the scientific depth and clarity of this section.
- Comment: As mentioned above, the draft has insufficient depth in discussing key literature. In fact, the authors have picked a limited number of trials and just summarized their objective outcomes, without interpreting them. Such numbers should be put in tables, and lucid explanations and conclusions must be made.
Response
- Thank you for this valuable feedback. In response, we have revised the manuscript to address the limited depth in discussing key literature. Specifically, we have included a more comprehensive selection of relevant clinical trials involving anlotinib, both as monotherapy and in combination with other therapies. The objective outcomes of these trials have been summarized in Table 1 for improved clarity and accessibility. In addition, we have provided more detailed narrative interpretations of the trial results, highlighting clinical significance, implications for practice, and observed trends. These enhancements aim to provide readers with a clearer understanding of the efficacy and potential of anlotinib-based regimens in advanced NSCLC.
- Comment: Likewise, the mechanisms of resistance and side effects of anlotinib are not discussed enough.
Response
- Thank you for your valuable comment. In response, we have substantially expanded the relevant sections to provide a more detailed discussion of both the mechanisms of resistance to anlotinib and its associated side effects in the context of NSCLC. The revised content now includes a comprehensive overview of resistance pathways—such as activation of alternative angiogenic signals, tumor microenvironment adaptations, and drug efflux mechanisms—as well as a detailed summary of commonly reported adverse events, including hypertension, hand-foot skin reaction, proteinuria, and thromboembolic risks. We have also supported these discussions with updated references to recent clinical and mechanistic studies. These revisions aim to provide greater depth and clinical relevance to the manuscript.
- Comment: Each abbreviation must be defined by its first appearance within the draft. This has been violated several times (e.g., p. 2, l. 60, and p. 2, l. 95).
Response
- We appreciate your observation regarding the inconsistent use of abbreviations. In response, we have carefully reviewed the entire manuscript to ensure that each abbreviation is clearly defined at its first appearance, including those noted on page 2, lines 60 and 95.
- Comment: Some parts of the draft need polishing to improve the readability.
For example, the following sentence can be shortened:
"Although there is a difference between the two groups in the OS, it is an insignificant difference (P = 0.2316)."
The following sentence, as another example, is hard to read (i.e., chemotherapy that contains):
"The approved first-line treatment paradigm for NSCLC is now chemotherapy that contains platinum, either in isolation or in conjunction with immunotherapeutic drugs."
Response
- We appreciate your feedback and agree that improving readability is important. In response, we have carefully revised the manuscript to streamline and polish several sections, including the sentence mentioned. We have shortened and clarified it to enhance flow and overall readability, ensuring that key information remains clear and concise.

Round 2
Reviewer 1 Report
Comments and Suggestions for Authors
Thanks for the responses to my comments. I agree that the manuscript can be accepted in its current format. Thanks.
Author Response
Response:
Thank you for your thoughtful review and your positive feedback. We are glad to hear that you are satisfied with the revisions and support the acceptance of the manuscript in its current form. We appreciate the time you took and your valuable insights throughout the review process.
Reviewer 3 Report
Comments and Suggestions for Authors
General Comments
I am generally satisfied with the authors' amendments. However, some awkward phrases in the newly added sections still need correction. Hence, I believe the draft can be accepted for publication after minor revision.
Minor issues
1. The English of some parts in the newly added sections must be rephrased. Some examples include:
- Bevacizumab just targets the VEGFR (p. 6, l. 246)
- Anlotinib has been investigated in the first-line context for NSCLC (p. 6, l. 265)
- for patients who shouldn’t receive (p. 7, l. 294)
- While, thromboembolic events are less common (p. 14, l. 535)
2. p. 6, l. 267: “available anti-angiogenic agents. 25 We postulated” Is this number a missed reference?
3. p. 6, l. 250: “anlotinib's exceptional anti-angiogenic properties, when compared to other TKIs”: using terms like exceptional is not consistent with the principles of scientific writing.
Author Response
Reviewer 3:
- Comment: I am generally satisfied with the authors' amendments. However, some awkward phrases in the newly added sections still need correction. Hence, I believe the draft can be accepted for publication after minor revision.
Response:
Thank you for your thoughtful review and your positive feedback. We appreciate your careful review and will revise the newly added sections to improve clarity and address any awkward phrasing. We’re grateful for your support in recommending acceptance pending minor revision.
- Comment: The English of some parts in the newly added sections must be rephrased. Some examples include:
- Bevacizumab just targets the VEGFR (p. 6, l. 246)
- Anlotinib has been investigated in the first-line context for NSCLC (p. 6, l. 265)
- for patients who shouldn’t receive (p. 7, l. 294)
- While, thromboembolic events are less common (p. 14, l. 535)
Response:
Thank you for your constructive feedback. We acknowledge that certain parts of the newly added sections require rephrasing for improved clarity and readability. We carefully revised these areas to ensure the language meets the required standards. We appreciate your guidance in identifying specific examples.
- Comment: 2. p. 6, l. 267: “available anti-angiogenic agents. 25 We postulated,” Is this number a missed reference?
Response:
Thank you for pointing this out. Yes, the number "25" appears to be a misplaced reference marker. We corrected this in the revised manuscript to ensure accuracy and clarity.
- Comment: 3. p. 6, l. 250: “anlotinib's exceptional anti-angiogenic properties, when compared to other TKIs”: Using terms like exceptional is not consistent with the principles of scientific writing.
Response:
Thank you for this valuable comment. We agree that the term “exceptional” is subjective and not appropriate for scientific writing. We revised the wording and deleted it.